# Training Deep AutoEncoders for Recommender Systems

## Abstract

This paper proposes a new model for the rating prediction task in recommender systems which significantly outperforms previous state-of-the art models on a time-split Netflix data set. Our model is based on deep autoencoder with 6 layers and is trained end-to-end without any layer-wise pre-training. We empirically demonstrate that: a) deep autoencoder models generalize much better than the shallow ones, b) non-linear activation functions with negative parts are crucial for training deep models, and c) heavy use of regularization techniques such as dropout is necessary to prevent overfitting. We also propose a new training algorithm based on iterative output re-feeding to overcome natural sparseness of collaborate filtering. The new algorithm significantly speeds up training and improves model performance. [1]

## 1 Introduction

Sites like Amazon, Netflix and Spotify use recommender systems to suggest items to users. Recommender systems can be divided into two categories: context-based and personalized recommendations.

Context based recommendations take into account contextual factors such as location, date and time (Adomavicius & Tuzhilin, 2011). Personalized recommendations typically suggest items to users using the collaborative filtering (CF) approach. In this approach the user's interests are predicted based on the analysis of tastes and preference of other users in the system and implicitly inferring "similarity" between them. The underlying assumption is that two people who have similar tastes, have a higher likelihood of having the same opinion on an item than two randomly chosen people.

In designing recommender systems, the goal is to improve the accuracy of predictions. The Netflix Prize contest provides the most famous example of this problem (Bennett et al., 2007): Netflix held the Netflix Prize to substantially improve the accuracy of the algorithm to predict user ratings for films. This is a classic CF problem: Infer the missing entries in an $mxn$ matrix, $R$, whose $(i, j)$ entry describes the ratings given by the $i$th user to the $j$th item. The performance is then measured using *Root Mean Squared Error (RMSE)*.

Training very deep autoencoders is non trivial both from optimization and regularization points of view. Early works on training auto-enocoders adapted layer-wise pre-training to solve optimization issues (Hinton & Salakhutdinov, 2006). In this work, we empirically show that optimization difficulties of training deep autoencoders can be solved by using *scaled exponential linear units (SELUs)*(Klambauer et al., 2017). This enables training without any layer-wise pre-training or residual connections. Since publicly available data sets for CF are relatively small, sufficiently large models can easily overfit. To prevent overfitting we employ heavy dropout with drop probability as high as 0.8. We also introduce a new output re-feeding training algorithm which helps to bypass the natural sparseness of updates in collaborative filtering and helps to further improve the model performance.

---

[1] Our code is available at `https://github.com/Anonymous`

## 1.1 RELATED WORK

Deep learning (LeCun et al., 2015) has led to breakthroughs in image recognition, natural language understanding, and reinforcement learning. Naturally, these successes fuel an interest for using deep learning in recommender systems. First attempts at using deep learning for recommender systems involved restricted Boltzman machines (RBM) (Salakhutdinov et al., 2007). Several recent approaches use autoencoders (Sedhain et al., 2015; Strub & Mary, 2015), feed-forward neural networks (He et al., 2017), neural autoregressive architectures (Zheng et al., 2016) and recurrent recommender networks (Wu et al., 2017). Many popular matrix factorization techniques can be thought of as a form of dimensionality reduction. It is, therefore, natural to adapt deep autoencoders for this task as well. *I-AutoRec* (item-based autoencoder) and *U-AutoRec* (user-based autoencoder) are first successful attempts to do so Sedhain et al. (2015). Stacked de-noising autoencoders has been sucesfully used on this task as well (Li et al., 2015; Wang et al., 2015).

There are many non deep learning types of approaches to collaborative filtering (CF) (Breese et al., 1998; Ricci et al., 2011). Matrix factorization techniques, such as alternating least squares (ALS) (Kim & Park, 2008; Koren et al., 2009) and probabilistic matrix factorization (Mnih & Salakhutdinov, 2008) are particularly popular. The most robust systems may incorporate several ideas together such as the winning solution to the Netflix Prize competition (Koren, 2009). Note that Netflix Prize data also includes temporal signal - time when each rating has been made. Thus, several classic CF approaches has been extended to incorporate temporal information such as TimeSVD++ Koren (2010), as well as more recent RNN-based techniques such as recurrent recommender networks Wu et al. (2017).

## 2 MODEL

Our model is inspired by *U-AutoRec* approach with several important distinctions. We train much deeper models. To enable this without any pre-training, we: a) use "scaled exponential linear units" (SELUs) Klambauer et al. (2017), b) use high dropout rates, and d) use iterative output re-feeding during training.

An autoencoder is a network which implements two transformations - encoder $encode(x) : R^n \rightarrow R^d$ and $decoder(z) : R^d \rightarrow R^n$. The "goal" of autoenoder is to obtain $d$ dimensional representation of data such that an error measure between $x$ and $f(x) = decode(encode(x))$ is minimized Hinton & Zemel (1994). Figure 1 depicts typical 4-layer autoencoder network. If noise is added to the data during encoding step, the autoencoder is called *de-noising*. Autoencoder is an excellent tool for dimensionality reduction and can be thought of as a strict generalization of principle component analysis (PCA) Hinton & Salakhutdinov (2006). An autoencoder without non-linear activations and only with "code" layer should be able to learn PCA transformation in the encoder if trained to optimize mean squared error (MSE) loss.

In our model, both encoder and decoder parts of the autoencoder consist of feed-forward neural networks with classical fully connected layers computing $l = f(W * x + b)$, where $f$ is some non-linear activation function. If range of the activation function is smaller than that of data, the last layer of the decoder should be kept linear. We found it to be very important for activation function $f$ in hidden layers to contain non-zero negative part, and we use SELU units in most of our experiments (see Section 3.2 for details).

If decoder mirrors encoder architecture (as it does in our model), then one can constrain decoder's weights $W_d^l$ to be equal to transposed encoder weights $W_e^l$ from the corresponding layer $l$. Such autoencoder is called *constrained* or *tied* and has almost two times less free parameters than unconstrained one.

**Forward pass and inference**. During forward pass (and inference) the model takes user represented by his vector of ratings from the training set $x \in R^n$, where $n$ is number of items. Note that $x$ is very sparse, while the output of the decoder, $f(x) \in R^n$ is dense and contains rating predictions for all items in the corpus.

## 2.1 Loss function

Since it doesn't make sense to predict zeros in user's representation vector $x$, we follow the approach from Sedhain et al. (2015) and optimize Masked Mean Squared Error loss:

$$MMSE = \frac{m_i * (r_i - y_i)^2}{\sum_{i=0}^{i=n} m_i} \tag{1}$$

where $r_i$ is actual rating, $y_i$ is reconstructed, or predicted rating, and $m_i$ is a mask function such that $m_i = 1$ if $r_i \neq 0$ else $m_i = 0$. Note that there is a straightforward relation between RMSE score and MMSE score: $RMSE = \sqrt{MMSE}$.

## 2.2 Dense re-feeding

During training and inference, an input $x \in R^n$ is very sparse because no user can realistically rate but a tiny fractions of all items. This poses problem for model training. Bayesian approches can be used to overcome this issue (Wang et al., 2015). On the other hand, autoencoder's output $f(x)$ is dense. Lets consider an idealized scenario with a *perfect f*. Then $f(x)_i = x_i, \forall i : x_i \neq 0$ and $f(x)_i$ accurately predicts all user's *future* ratings for items $i : x_i = 0$. This means that if user rates new item $k$ (thereby creating a new vector $x'$) then $f(x)_k = x'_k$ and $f(x) = f(x')$. Hence, in this idealized scenario, $y = f(x)$ should be a *fixed point* of a well trained autoencoder: $f(y) = y$.

To explicitly enforce fixed-point constraint and to be able to perform dense training updates, we augment every optimization iteration with an iterative dense re-feeding steps (3 and 4 below) as follows:

1. Given sparse $x$, compute dense $f(x)$ and loss using equation 1 (forward pass)
2. Compute gradients and perform weight update (backward pass)
3. Treat $f(x)$ as a new example and compute $f(f(x))$. Now both $f(x)$ and $f(f(x))$ are dense and the loss from equation 1 has all $m$ as non-zeros. (second forward pass)
4. Compute gradients and perform weight update (second backward pass)

Steps (3) and (4) can be also performed more than once for every iteration.

## 3 Experiments and Results

### 3.1 Experiment setup

For the rating prediction task, it is often most relevant to predict *future* ratings given the *past* ones instead of predicting ratings missing at random. For evaluation purposes we followed Wu et al. (2017) exactly by splitting the original Netflix Prize Bennett et al. (2007) training set into several training and testing intervals based on time. Training interval contains ratings which came in earlier than the ones from testing interval. Testing interval is then randomly split into Test and Validation subsets so that each rating from testing interval has a 50% chance of appearing in either subset. Users and items that do not appear in the training set are removed from both test and validation subsets. Table 1 provides details on the data sets.[2]

For most of our experiments we uses a batch size of 128, trained using SGD with momentum of 0.9 and learning rate of 0.001. We used xavier initialization to initialize parameters. Note, that unlike Strub & Mary (2015) we did not use any layer-wise pre-training. We believe that we were able to do so successfully because of choosing the right activation function (see Section 3.2).

### 3.2 Effects of the activation types

To explore the effects of using different activation functions, we tested some of the most popular choices in deep learning : sigmoid, "rectified linear units" (RELU), $max(relu(x), 6)$ or RELU6,

---

[2]Note, that while checking our data set statistics with first author of (Wu et al., 2017) it was determined that their publication contained the following typo: "Netflix 6m" should be "Netflix 3m".

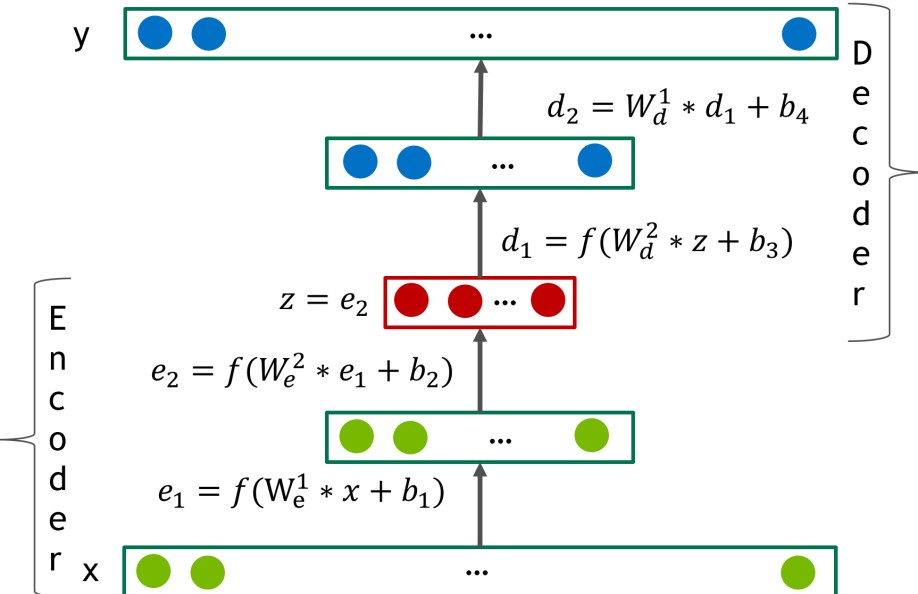

Figure 1: AutoEncoder consists of two neural networks, encoder and decoder, fused together on the "representation" layer $z$. Encoder has 2 layers $e_1$ and $e_2$ and decoder has 2 layers $d_1$ and $d_2$. Dropout may be applied to coding layer $z$.

Table 1: Subsets of Netflix Prize training set used in our experiments. We made sure that these splits match exactly the ones used in (Wu et al., 2017).

|  | Full | 3 months | 6 months | 1 year |
|---|---|---|---|---|
| Training | 12/99-11/05 | 09/05-11/05 | 06/05-11/05 | 06/04-05/05 |
| Users | 477,412 | 311,315 | 390,795 | 345,855 |
| Ratings | 98,074,901 | 13,675,402 | 29,179,009 | 41,451,832 |
| Testing | 12/05 | 12/05 | 12/05 | 06/05 |
| Users | 173,482 | 160,906 | 169,541 | 197,951 |
| Ratings | 2,250,481 | 2,082,559 | 2,175,535 | 3,888,684 |

hyperbolic tangent (TANH), "exponential linear units" (ELU) (Clevert et al., 2015), leaky relu (LRELU) (Xu et al., 2015) , "self-gated activation function" (SWISH) (Ramachandran et al., 2017), and "scaled exponential linear units" (Klambauer et al., 2017) (SELU) on the 4 layer autoencoder with 128 units in each hidden layer. Because ratings are on the scale from 1 to 5, we keep last layer of the decoder linear for sigmoid and tanh-based models. In all other models activation function is applied in all layers.

We found that on this task ELU, SELU and LRELU perform much better than SIGMOID, RELU, RELU6, TANH and SWISH. Figure 2 clearly demonstrates this. There are two properties which seems to separate activations which perform well from those which do not: a) non-zero negative part and b) unbounded positive part. Hence, we conclude, that in this setting these properties are important for successful training. Thus, we use SELU activation units and tune SELU-based networks for performance.

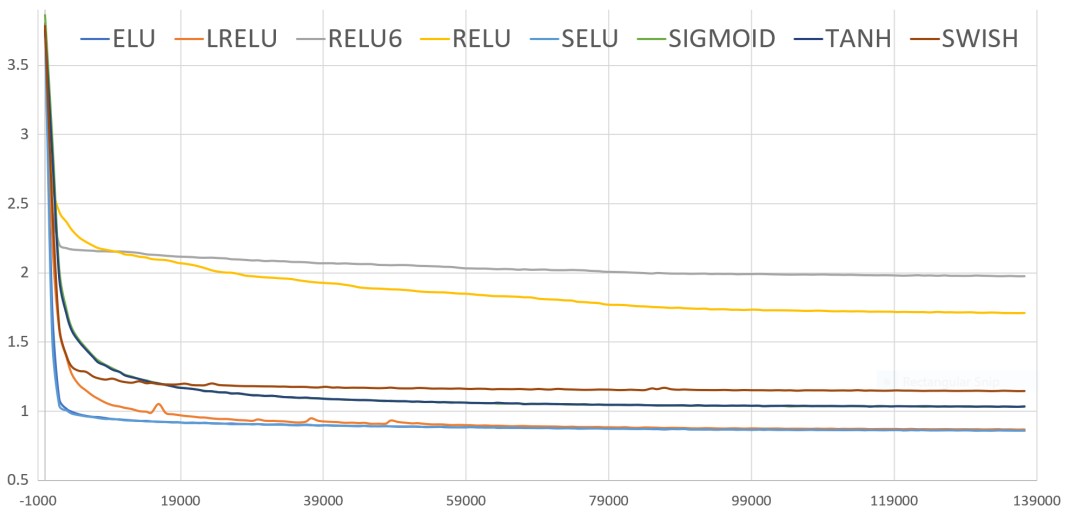

Figure 2: Training RMSE per mini-batch. All lines correspond to 4-layers autoencoder (2 layer encoder and 2 layer decoder) with hidden unit dimensions of 128. Different line colors correspond to different activation functions. TANH and SIGMOID lines are very similar as well as lines for ELU and SELU. The best performing activation functions are ELU and SELU.

### 3.3 OVER-FITTING THE DATA

The largest data set we use for training, "Netflix Full" from Table 1, contains 98M ratings given by 477K users. Number of movies (e.g. items) in this set is $n = 17,768$. Therefore, the first layer of encoder will have $d * n + d$ weights, where $d$ is number of units in the layer.

For modern deep learning algorithms and hardware this is relatively small task. If we start with single layer encoders and decoders we can quickly overfit to the training data even for $d$ as small as 512. Figure 3 clearly demonstrates this. Switching from unconstrained autoencoder to constrained reduces over-fitting, but does not completely solve the problem.

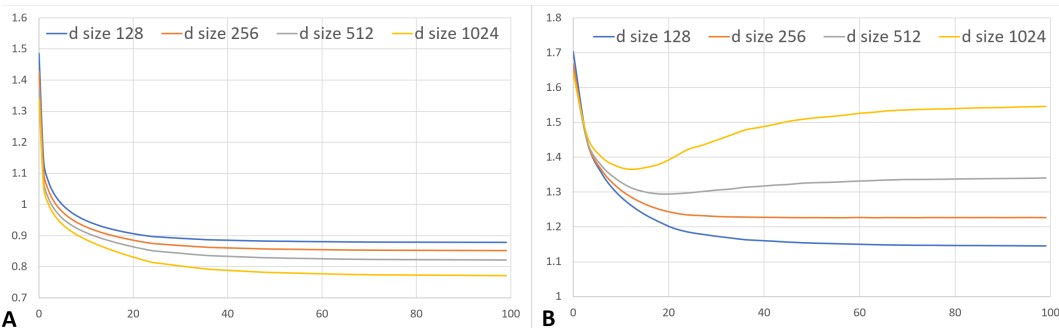

Figure 3: Single layer autoencoder with 128, 256, 512 and 1024 hidden units in the coding layer. A: training RMSE per epoch; B: evaluation RMSE per epoch.

### 3.4 GOING DEEPER

While making layers wider helps bring training loss down, adding more layers is often correlated with a network's ability to generalize. In this set of experiments we show that this is indeed the case here. We choose small enough dimensionality ($d = 128$) for all hidden layers to easily avoid over-fitting and start adding more layers. Table 2 shows that there is a positive correlation between the number of layers and the evaluation accuracy.

Table 2: Depth helps generalization. Evaluation RMSE of the models with different number of layers. In all cases the hidden layer dimension is 128.

| Number of layers | Evaluation RMSE | params |
|---|---|---|
| 2 | 1.146 | 4,566,504 |
| 4 | 0.9615 | 4,599,528 |
| 6 | 0.9378 | 4,632,552 |
| 8 | 0.9364 | 4,665,576 |
| 10 | 0.9340 | 4,698,600 |
| 12 | 0.9328 | 4,731,624 |

Going from one layer in encoder and decoder to three layers in both provides good improvement in evaluation RMSE (from 1.146 to 0.9378). After that, blindly adding more layers does help, however it provides diminishing returns. Note that the model with single $d = 256$ layer in encoder and decoder has 9,115,240 parameters which is almost two times more than any of these deep models while having much worse evauation RMSE (above 1.0).

### 3.5  DROPOUT

Section 3.4 shows us that adding too many small layers eventually hits diminishing returns. Thus, we start experimenting with model architecture and hyper-parameters more broadly. Our most promising model has the following architecture: $n, 512, 512, 1024, 512, 512, n$, which means 3 layers in encoder (512,512,1024), coding layer of 1024 and 3 layers in decoder of size 512,512,n. This model, however, quickly over-fits if trained with no regularization. To regularize it, we tried several dropout values and, interestingly, very high values of drop probability (e.g. 0.8) turned out to be the best. See Figure 4 for evaluation RMSE. We apply dropout on the encoder output only, e.g. $f(x) = decode(dropout(encode(x)))$. We tried applying dropout after every layer of the model but that stifled training convergence and did not improve generalization.

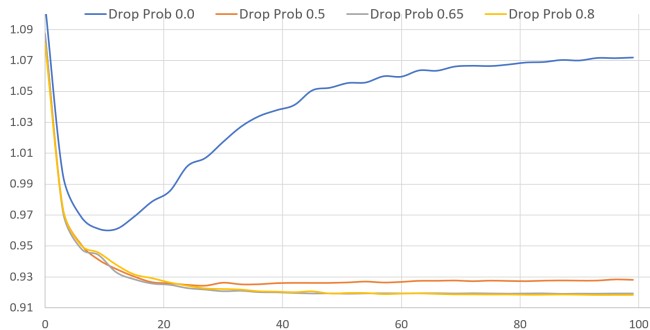

Figure 4: Effects of dropout. Y-axis: evaluation RMSE, X-axis: epoch number. Model with no dropout (Drop Prob 0.0) clearly over-fits. Model with drop probability of 0.5 over-fits as well (but much slowly). Models with drop probabilities of 0.65 and 0.8 result in RMSEs of 0.9192 and 0.9183 correspondingly.

### 3.6  DENSE RE-FEEDING

Iterative dense re-feeding (see Section 2.2) provides us with additional improvement in evaluation accuracy for our 6-layer-model: $n, 512, 512, 1024, dp(0.8), 512, 512, n$ (referred to as Baseline below). Here each parameter denotes the number of inputs, hidden units, or outputs and $dp(0.8)$ is a dropout layer with a drop probability of 0.8. Just applying output re-feeding did not have significant

Table 3: Test RMSE of different models. I-AR, U-AR and RRN numbers are taken from (Wu et al., 2017)

| DataSet | I-AR | U-AR | RRN | DeepRec |
|---|---|---|---|---|
| Netflix 3 months | 0.9778 | 0.9836 | 0.9427 | **0.9373** |
| Netfix Full | 0.9364 | 0.9647 | 0.9224 | **0.9099** |

impact on the model performance. However, *in conjunction with the higher learning rate*, it did significantly increase the model performance. Note, that with this higher learning rate (0.005) but without dense re-feeding, the model started to diverge. See Figure 5 for details.

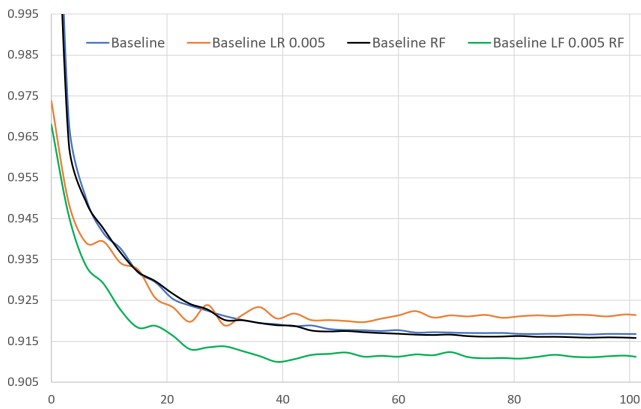

Figure 5: Effects of dense re-feeding. Y-axis: evaluation RMSE, X-axis: epoch number. Baseline model was trained with learning rate of 0.001. Applying re-feeding step with the same learning rate almost did not help (Baseline RF). Learning rate of 0.005 (Baseline LR 0.005) is too big for baseline model without re-feeding. However, increasing both learning rate and applying re-feeding step clearly helps (Baseline LR 0.005 RF).

Applying dense re-feeding and increasing the learning rate, allowed us to further improve the evaluation RMSE from 0.9167 to 0.9100. Picking a checkpoint with best *evaluation RMSE* and computing *test* RMSE gives as **0.9099**, which we believe is significantly better than other methods.

### 3.7 COMPARISON WITH OTHER METHODS

We compare our best model with Recurrent Recommender Network from Wu et al. (2017) which has been shown to outperform PMF (Mnih & Salakhutdinov, 2008), T-SVD (Koren, 2010) and I/U-AR (Sedhain et al., 2015) on the data we use (see Table 1 for data description). Note, that unlike T-SVD and RRN, our method does not explicitly take into account temporal dynamics of ratings. Yet, Table 3 shows that it is still capable of outperforming these methods on *future* rating prediction task. We train each model using only the training set and compute evaluation RMSE for 100 epochs. Then the checkpoint with the highest evaluation RMSE is tested on the test set.

"Netflix 3 months" has 7 times less training data compared to "Netflix full", it is therefore, not surprising that the model's performance is significantly worse if trained on this data alone (0.9373 vs 0.9099). In fact, the model that performs best on "Netflix full" over-fits on this set, and we had to reduce the model's complexity accordingly (see Table 4 for details).

Table 4: Test RMSE achieved by DeepRec on different Netflix subsets. All models are trained with one iterative output re-feeding step per each iteration.

| DataSet | RMSE | Model Architecture |
|---|---|---|
| Netflix 3 months | **0.9373** | $n, 128, 256, 256, dp(0.65), 256, 128, n$ |
| Netflix 6 months | **0.9207** | $n, 256, 256, 512, dp(0.8), 256, 256, n$ |
| Netflix 1 year | **0.9225** | $n, 256, 256, 512, dp(0.8), 256, 256, n$ |
| Netfix Full | **0.9099** | $n, 512, 512, 1024, dp(0.8), 512, 512, n$ |

## 4 CONCLUSION

Deep learning has revolutionized many areas of machine learning, and it is poised do so with recommender systems as well. In this paper we demonstrated how very deep autoencoders can be successfully trained even on relatively small amounts of data by using both well established (dropout) and relatively recent ("scaled exponential linear units") deep learning techniques. Further, we introduced *iterative output re-feeding* - a technique which allowed us to perform dense updates in collaborative filtering, increase learning rate and further improve generalization performance of our model. On the task of *future* rating prediction, our model outperforms other approaches even without using additional temporal signals.

While our code supports item-based model (such as *I-AutoRec*) we argue that this approach is less practical than user-based model (*U-AutoRec*). This is because in real-world recommender systems, there are usually much more users then items. Finally, when building personalized recommender system and faced with scaling problems, it can be acceptable to sample items but not users.

### ACKNOWLEDGMENTS

We thank the author of (Wu et al., 2017), Chao-Yuan Wu, for fruitfull discussion and help validating our data sets.

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
