# OpenReview forum: "Training Deep AutoEncoders for Recommender Systems"
_ICLR.cc/2018/Conference — Reject_

### Official Review · AnonReviewer2 · 2017-11-25
**Lacks novelty and please no more rating predictions**

**Rating:** 4
**Confidence:** 5

**Review:**

This paper presents a deep autoencoder model for rating prediction. The autoencoder takes the user’s rating over all the items as input and tries to predict the observed ratings in the output with mean squared error. A few techniques are applied to make the training feasible without layer-wise pre-training: 1) SELU activation. 2) dropout with high probability. 3) dense output re-feeding. On the Netflix prize dataset, the proposed deep autoencoder outperforms other state-of-the-art approaches.

Overall, the paper is easy to follow. However, I have three major concerns regarding the paper that makes me decide to reject it.

1. Lack of novelty. The paper is essentially a deeper version of the U-AutoRec (Sedhain et al. 2015) with a few recently emerged innovations in deep learning. The dense output re-feeding is not something particularly novel, it is more or less a data-imputation procedure with expectation-maximization — in fact if the authors intend to seek explanation for this output re-feeding technique, EM might be one of the interpretations. And similar technique (more theoretically grounded) has been applied in image imputation for variational autoencoder (Rezende et al. 2014, Stochastic Backpropagation and Approximate Inference in Deep Generative Models).

2. The experimental setup is also worth questioning. Using a time-split dataset is of course more challenging. However, the underlying assumption of autoencoders (or more generally, latent factor models like matrix factorization) is the all the ratings are exchangeable (conditionally independent given the latent representations), i.e., autoencoders/MF are not capable of inferring the temporal information from the data, Thus it is not even a head-to-head comparison with a temporal model (e.g., RNN in Wu et al. 2017). Of course you can still apply a static autoencoder to time-split data, but what ends up happening is the model will use its capacity to try to explain the temporal signal in the data — a deeper model certainly has more extra capacity to do so. I would suggest the authors comparing on a non-time-split dataset with other static models, like I(U)-AutoRec/MF/CF-NADE (Zheng et al. 2016)/etc.

3. Training deep models on recommender systems data is impressive. However, I would like to suggest we, as a community, start to step away from the task of rating predictions as much as we can, especially in more machine-learning-oriented venues (NIPS, ICML, ICLR, etc.) where the reviewers might be less aware of the shift in recommender systems research. (The task of rating predictions was made popular mostly due to the Netflix prize, yet even Netflix itself has already moved on from ratings.) Training (and evaluating) with RMSE on the observed ratings assumes all the missing ratings are missing at random, which is clearly far from realistic for recommender systems (see Marlin et al. 2007, Collaborative Filtering and the Missing at Random Assumption). In fact, understanding why some of the ratings are missing presents a unique challenge for the recommender systems. See, e.g., Steck 2010, Training and testing of recommender systems on data missing not at random, Liang et al. 2016, Modeling user exposure in recommendation, Schnabel et al. 2016, Recommendations as Treatments: Debiasing Learning and Evaluation. A model with good RMSE in a lot of cases does not translate to good recommendations (Cremonesi et al. 2010, Performance of recommender algorithms on top-n recommendation tasks
). As a first step, at least start to use all the 0’s in the form of implicit feedback and focus on ranking-based metrics other than RMSE.

---

> ### Author Response · Authors · 2017-12-24
> **Thank you for reviewing our work.**
>
> Please see below our responses to your concerns.
>
> Concern 1. Yes, the model is a deeper version of U-AutoRec. But simply stacking more layers does not always work and we show that in this case the following changes were needed: a) new activation functions and dropouts, and b) new optimization scheme (dense re-feeding). Most importantly, we show how *each* change impacts performance and enables training of deeper and deeper models - which we think is of interest to the ICLR audience.
> The EM-based point of view on dense re-feeding is an interesting angle, thanks for pointing this out.
>
> Concern 2. We strongly disagree that experimental setup is questionable.
>
> As you commented yourself, "Using a time-split dataset is of course more challenging" - this is true. However, in practice, we are interested in predicting *future* ratings/interests, given the past ones. Therefore, time-based benchmark makes much more sense then random-based one.
>
> We also made sure (by corresponding with Wu et al. 2017 authors) that our dataset splits match theirs exactly.
> Yes, we agree that our model does not model explicitly temporal signal (unlike RRN from Wu et al.) which makes it even more interesting that our model beats RRN which is RNN-based and explicitly takes time as a signal.
> Perhaps, you are right that "model will use its capacity to try to explain the temporal signal in the data", however we do not see how this makes experimental setup questionable.
>
> Concern 3.
>
> We agree that, in practice, for the *production* recommender system, ratings prediction task is not particularly valuable due to many reasons including the ones that you've cited above. In fact, from our experience, production recommender systems are more similar to search engines in the sense that they take myriads of signals into the account with ratings data being just one of them.
>
> Nevertheless, often a simple test (Netflix) and metric (RMSE) which could tell whether algo 1 models rating data potentially better than algo 2 is desirable.
>
> In particular, we think that for ICLR audience it would be interesting to see how classical well known machine-learning techniques such as matrix factorization can be replaced by a deep learning based model without going too deeply into the specifics of the domain of application area (that would be more appropriate for a RecSys paper).

---

### Official Review · AnonReviewer3 · 2017-11-28
**The paper is more like a technical report rather than a research paper**

**Rating:** 3
**Confidence:** 4

**Review:**

This paper proposed to use deep AE to do rating prediction tasks in recommender systems.
Some of the conclusions of the paper, e.g. deep models perform bettern than shallow ones, the non-linear activation
function is important, dropout is necessary to prevent overfitting, are well known, and hence is of less novelty.
The proposed re-feeding algorithm to overcome natural sparseness of CF is interesting, however, I don't think it is enough to support being accepted by ICLR.
Some reference about rating prediction are missing, such as "A neural autoregressive approach to collaborative filtering, ICML2016". And it would be better to show the performance of the model on implicit rating data, since it is more desirable in practice, since many industry applications have only implicit rating (e.g. whether the user watches the movie or not.).

---

> ### Author Response · Authors · 2017-12-24
> **Thank you for reviewing our work.**
>
> Stacking many layers together does not always work and often well-known techniques (more recent activation functions, dropouts, etc.) need to be thoughtfully combined to successfully train deeper models. This is why the paper reads as a technical report - we evaluated the effect of every "trick" we used. We think this, as well as new optimization scheme (dense re-feeding) may be of interest to the ICLR audience.
>
> The reference to “A neural autoregressive approach to collaborative filtering, ICML2016" is indeed relevant to our work and we added it into the “Related work” section.

---

### Official Review · AnonReviewer1 · 2017-11-28
**Thorough experimental paper improving Netflix RMSE.**

**Rating:** 6
**Confidence:** 4

**Review:**

In this paper the authors present a model for more accurate Netflix recommendations (rating predictions, RMSE).  In particular, the authors demonstrate that a deep autoencoder, carefully tuned, can out-perform  more complex RNN-based models that have temporal information.  The authors examine how different non-linear activations, model size, dropout, and a novel technique called "dense re-feeding" can together improve DNN-based collaborative filtering.

Pros:
- The accuracy results are impressive and a useful datapoint in how to build a DNN-based recommender.
- The dense re-feeding technique seems to be novel with incremental (but meaningful) benefits.

Cons:
- Experimental results on only one dataset.
- Difficult to know if the results are generalizable.

---

> ### Author Response · Authors · 2017-12-24
> **Thank you for reviewing our work.**
>
> Similarly to recent approaches, we performed several different time-splits of Neflix data to see if the results will generalize.
> Netflix data was chosen, because it is the largest publicly available ratings data which would also help to show how scalable our approach is (takes few hours to train on single GPU).
>
> We do agree that more experiments would make model and results stronger and we are currently working on getting access to a bigger datasets (which is not public unfortunately).

---

### Public Comment · (anonymous) · 2017-11-19
**As far as I know, baselines such as PMF (without biased term) performs very poorly in rating prediction task. Biased matrix factorization is a very solid rating prediction baseline.**

It can be easily verified by using open source toolkit, such as librec, MyMediaLite, etc.

---

### Decision · Program_Chairs · 2018-01-29
**ICLR 2018 Conference Acceptance Decision**

**Decision:**

Reject

**Comment:**

meta score: 4

The paper uses a deep autoencoder to rating prediction, with experiments on netflix.

Pros
 - Proposed dense refeeding approach appears novel
 - Good experimental results

Cons
 - limited experimentation
 - main novelty (dense refeeding) is not well linked to existing data imputation approaches
 - novel contribution is otherwise quite limited